## [Decision Letter · Decision Letter 0]

12 May 2021

HAI-1 is an Independent Predictor of Lung Cancer Mortality and is Required for M1 Macrophage Polarization

PONE-D-21-10515

Dear Dr. Borowicz,

We’re pleased to inform you that your manuscript has been judged scientifically suitable for publication and will be formally accepted for publication once it meets all outstanding technical requirements.

Kind regards,

Rajeev Samant

Academic Editor

PLOS ONE

Additional Editor Comments:

This is a remarkably well presented set of rigorous observations.

1. In your Methods section, please provide additional details regarding the "lung cancer and adjacent normal tissues" used in your study and ensure you have described the source. For more information regarding PLOS' policy on materials sharing and reporting, see https://journals.plos.org/plosone/s/materials-and-software-sharing#loc-sharing-materials.

Reviewers' comments:

Reviewer's Responses to Questions

**Comments to the Author**

1. Is the manuscript technically sound, and do the data support the conclusions?

Reviewer #1: Yes

2. Has the statistical analysis been performed appropriately and rigorously? 

Reviewer #1: Yes

3. Have the authors made all data underlying the findings in their manuscript fully available?

Reviewer #1: Yes

4. Is the manuscript presented in an intelligible fashion and written in standard English?

Reviewer #1: Yes

5. Review Comments to the Author

Reviewer #1: This is a very well conducted and written work, showing the impact of HGF activation/inhibition by HAI-1 and an associated increase in the density of M2 macrophages. I recommend it for publication in the present format.

6. PLOS authors have the option to publish the peer review history of their article (what does this mean?). If published, this will include your full peer review and any attached files.

Reviewer #1: No

---

## [Editor Report · Acceptance letter]

19 May 2021

PONE-D-21-10515 

HAI-1 is an Independent Predictor of Lung Cancer Mortality and is Required for M1 Macrophage Polarization 

Dear Dr. Borowicz:

I'm pleased to inform you that your manuscript has been deemed suitable for publication in PLOS ONE. Congratulations! Your manuscript is now with our production department. 

Kind regards, 

on behalf of

Dr. Rajeev Samant 

Academic Editor

PLOS ONE